# Immunogenicity of Different Types of Adjuvants and Nano-Adjuvants in Veterinary Vaccines: A Comprehensive Review

**DOI:** 10.3390/vaccines11020453

**Published:** 2023-02-16

**Authors:** Soren Nooraei, Alireza Sarkar Lotfabadi, Milad Akbarzadehmoallemkolaei, Nima Rezaei

**Affiliations:** 1Faculty of Veterinary Medicine, Shahrekord University, Shahrekord 8818634141, Iran; 2Animal Model Integrated Network (AMIN), Universal Scientific Education and Research Network (USERN), Tehran 1419733151, Iran; 3Research Center for Immunodeficiencies, Children’s Medical Center, Tehran University of Medical Sciences, Dr. Gharib St, Keshavarz Blvd, Tehran 1419733151, Iran; 4Department of Immunology, School of Medicine, Tehran University of Medical Sciences, Tehran 1417653761, Iran

**Keywords:** veterinary vaccines, adjuvant, nano-adjuvant, immunogenicity

## Abstract

Vaccination is the best way to prevent and reduce the damage caused by infectious diseases in animals and humans. So, several vaccines are used for prophylactic purposes before the pathogen infects, while therapeutic vaccines strengthen the immune system after infection with the pathogen. Adjuvants are molecules, compounds, or macromolecules that enhance non-specific immunity and, in collaboration with antigen(s), can improve the body’s immune responses and change the type of immune response. The potential and toxicity of adjuvants must be balanced to provide the safest stimulation with the fewest side effects. In order to overcome the limitations of adjuvants and the effective and controlled delivery of antigens, attention has been drawn to nano-carriers that can be a promising platform for better presenting and stimulating the immune system. Some studies show that nanoparticles have a more remarkable ability to act as adjuvants than microparticles. Because nano-adjuvants inactively target antigen-presenting cells (APCs) and change their chemical surface, nanoparticles also perform better in targeted antigen delivery because they cross biological barriers more easily. We collected and reviewed various types of nano-adjuvants with their specific roles in immunogenicity as a prominent strategy used in veterinary vaccines in this paper.

## 1. Introduction

In its original sense, the purpose of vaccination is to mimic the natural immunity of non-pathogenic components and establish a robust immune response and long-term protection against infection. However, the use of pathogenic components or organisms close to them is still in question [1,2].

Vaccine production has historically been recognized as one of the most successful public health experiences. Vaccine advances owe much to the “isolate, inactivate, and inject Louis Pasteur” model [3,4]. Edward Jenner first noticed about a hundred years earlier that cowpox infection made people immune to smallpox infection. In the late 1800s, Louis Pasteur made the first classical vaccines, creating what we now call vaccination [5]. For the first time, in 1881, Louis Pasteur used the word vaccine for immunogens that were used for diseases other than smallpox. Pasteur’s research showed that it was possible to weaken or inactivate microbes. Studies on fowl cholera and anthrax led to the concepts of chemical inactivation as a means of reducing the pathogenicity of microorganisms. In the rabies studies, they investigated an alternative strategy to reduce or eliminate the pathogenicity of serial passage in animals (lapinization or passage of rabbits) or other animal-derived tissues. These studies resulted in the successful control of anthrax and especially rabies [6]. Research by Salmon and Smith (1886) clearly showed that some microbes could be completely inactivated (killed), which led to successful immunization against typhoid fever, tuberculosis, rinderpest, and foot and mouth disease (FMD) [7]. Gaston Ramon at the Pasteur Institute developed the principles of weakening and inactivating microbial toxins. To create anatoxin, a tetanus toxin was inactivated by heat and formalin in 1924 which showed a better effect with aluminum hydroxide adjuvant. In the early 20th century, these advances in process and formulation produced horse sera with anti-diphtheria and Tetanus toxin neutralizing antibodies were developed and refined for preventive use [6].

In the mid-1950s, veterinarians routinely used brain tissue-derived rabies vaccines in dogs. The main biological products used at that time included rabies vaccine, canine distemper virus/hepatitis vaccine and antisera, hog cholera vaccines and antisera, leptospirosis bacteriens, and clostridial toxins. With the development and increase of production capacity over time, the vaccination of companion animals was expanded and rabies vaccines for cats, feline herpes virus, parvovirus in cats and dogs and calcivirus in cats were produced. The term vaccine is now used to describe many therapeutic or prophylactic formulations and products that stimulate active immunity in the vaccinated animal [6].

According to studies, vaccination is the best way to prevent and reduce the damage caused by infectious diseases in animals and humans [8]. Vaccines are recognized as one of the most essential and practical public health indicators. In the last century, they have saved countless lives, so they are considered for treating and preventing diseases. In addition, the World Health Organization recommends vaccination to prevent diseases [9].

Several vaccines are used for prophylactic purposes before the disease to protect the body against possible exposure to the pathogen, while therapeutic vaccines strengthen the immune system after infection with the pathogen. Antigens from vaccines used for prevention are usually taken by antigen-presenting cells (APCs), such as dendritic cells (DCs), and then prepared and processed. These cells migrate to the lymph nodes after puberty and then present antigens to B cells, CD4 and CD8, to detect possible infectious bodies. Therapeutic vaccines have been developed and designed to enhance tumor antigen delivery and cellular immune responses against tumors. In this regard, adjuvants are injected at the initial site of the tumor to promote the accumulation of DCs. As DCs mature and differentiate, tumor antigens are delivered to CD4 and CD8 cells. In the next step, the number of CD8T and natural killer cells (NK cells) increases to identify cells at the primary, metastatic, and distant cancer stages in the body [10,11].

Vaccines are classified into two types: live attenuated and inactivated. Inactivated vaccines contain pathogens that chemical treatments have killed. They are safer and more stable than live attenuated vaccines (LAVs) and cannot reactivate, even in immunosuppressed patients. However, they are less effective in inducing immunity than LAVs and usually require multiple doses to induce a humoral response, which is often not permanent. Current examples of vaccines in this category include those for polio (i.e., inactivated polio vaccine-IPV), hepatitis A, rabies, and influenza [5].

Immune systems can detect the causative agents of living diseases, constantly trigger a specific immune response, and help the immune system acquire immune memory because they behave similarly to natural infectious carriers. The most serious concern with using this type of vaccine is the possibility of reverting to a pathogenic state. The delay in immunogenicity or induction of abortion in the host receiving the vaccine is among the problems of using this type of vaccine. On the other hand, inactivated vaccines are safe, do not have pathogenic problems, and are easier to produce. Nevertheless, in most cases, several stages of immunization are required for long-term safety, and we need a large amount of antigen. It may also fail to provide local safety and to adequately activate all safety agents, although a meaningful way to improve their performance is to use adjuvants [12,13]. Adjuvants correct the poor immunogenicity of inactivated and subunit-based vaccines. These compounds significantly increase the vaccine’s immunogenicity by acting as an immunomodulatory and delivery system [5].

Vaccines can be administered by injectable, oral, or inhaled methods. Due to its non-invasive nature, oral administration has no pain or discomfort. However, oral vaccines should be resistant to the pH conditions of the gastrointestinal tract, and, of course, their bioavailability is not reduced. Nowadays, most vaccines are injected because they are more effective and practical [14,15,16].

Today, some of the most widely used veterinary-licensed vaccines to control livestock infectious diseases use Edward Jenner’s experimental technology. Jenner used the live attenuated virus to create immunity against smallpox, which he called the vaccine. After Jenner, Louis Pasteur invented another way to make a vaccine by using live microorganisms. In general, vaccines in veterinary medicine increase production and protect people who work in veterinary medicine [17].

When vaccines are used optimally, they can prevent the manifestations of diseases, reduce the transmission of diseases, reduce the need for medication, and improve the health and well-being of animals. They can prevent the transmission of diseases that are common between humans and livestock to ensure human health indirectly [18]. There are challenges in creating an optimal vaccination program, one of which is dealing with the great diversity of animals because there is no optimal unified program for vaccination. Moreover, the use of vaccines in animal health is not limited to the protection of animals, and it is mentioned in public health programs as one of the main elements of public health. With the availability of conditions, management modeling strategies and possible plans will be related to such things, such as (I) (DIVA) protective vaccines against common human diseases, (II) an effective predictive model, and (III) enforceable policy measures to prevent and control dangerous human and animal diseases in local, state, and national areas [19,20].

Producing vaccines in veterinary medicine has advantages and disadvantages compared to the production of human vaccines, such as smaller market sizes and lower selling prices. At the same time, there is greater complexity and range among hosts and pathogens. Therefore, less investment is made in the research and development of animal vaccines. For example, the market value of the most successful animal health vaccine ever produced against foot-and-mouth disease (FMD) is 10–20% of the market value of the papillomavirus cervical cancer vaccine in humans. However, there is less pre-clinical monitoring of the production of veterinary vaccines. Veterinary scientists can do research sooner than human vaccine providers. Therefore, the return on investment in this field and the time to launch their sales are shorter [2].

The next step in vaccine development is the use of engineered nanoparticles (NPs) that serve as vaccine delivery platforms that can protect the antigens of the vaccine and, at the same time, are innovative adjuvants that can fine-tune immunity. In addition to their function, NPs show an intrinsic adjuvant activity that, after being internalized by APCs, allows NPs to form a complex that induces and absorbs immune cells [5,21].

## 2. Adjuvant

### 2.1. What Is an Adjuvant

The word adjuvant has a Greek root, and it is derived from ADJUVARE. Adjuvants could be molecules, compounds, or macromolecules [17]. Adjuvants enhance non-specific immunity, and in collaboration with antigen(s), they can improve the body’s immune responses and change the type of immune response [22]. Today, there is an essential need for veterinary and human adjuvants in vaccines. Recently, efforts have been made to reduce the complexity of antigens, such as pure antigens, recombinants, synthetic proteins, or even synthetic peptides that require adjuvants, instead of using the whole inactive organism without adjuvants to induce effective immunity [17]. However, after the use of adjuvants, several immune-stimulating properties and systemic reactions have been observed in humans, which has led to the use of a small number of adjuvants in human vaccines [23].

Unfortunately, despite much research on adjuvants, the mechanism of action of all adjuvants has not been accurately identified. However, based on the evidence obtained, it generally seems that they increase the supply and stability of antigens or act as modulators of the immune system. In some cases, it has been observed that an adjuvant can have more than one mechanism of action. These methods have often successfully created immunity based on antibodies [24].

### 2.2. Role of Adjuvants in the Body

The purpose of using adjuvants is to improve the immune response to adjuvants, which can be effective in several ways, as shown in Figure 1 [1].

An ideal adjuvant should have the following characteristics of biological activity: The formulation should be safe and effective for all ages. Immune responses should be increased in very young, elderly, or immunocompromised populations. Moreover, more metabolizable compounds should be considered. Adjuvant activity should be regional and transient, and adjuvants should not directly affect lymphocytes and should not be associated with non-specific B and T cellular responses. In adjuvant immunization, different methods require different formulations, and at the same time, the adjuvant must reduce the required antigen dose or the number of reminders needed. The range of adjuvants’ responses to the pathogen should be broad, while neutralizing antibody responses should be increased by adjuvants. The adjuvant should induce or prolong CD4+ or CD8+ cell responses to cell-mediated immunity. The quality of the adjuvant’s immune response should be able to shape the immune response, e.g., the equilibrium of Th1 versus Th2. Synthetic adjuvants are more important than others because of their purity, tolerability, and safety. Plant adjuvants are acceptable when synthetic types are costly or less productive to produce. At the same time, animal sources are concerned about relationships with diseases and should not be used in general; metabolizable or excretable compounds are more important. Active drug components must be stable and retain their chemical structure, particle size, shape, and appearance, as well as conditions, such as packaging under vacuum, being stable for years against oxidative decomposition [25].

### 2.3. Different Types of Adjuvants

Adjuvants may have different uses, and they are generally divided into three groups: delivery systems, immune-boosting molecules, or a combination of the two. Immune-boosting molecules that act as ligands for innate immune responses, including TLRs, NLRs (NOD-like receptors), lectin type-C, and I-RIG receptors, can directly activate innate immune responses. The mechanism of action of other immune-boosting molecules such as QS21 and other saponins is unknown. The most important immune-boosting molecules are the MPL ligand TLR4 and the CPG ligand TLR9. In delivery systems, immobilized safety compounds are used, and their purpose is the effective transfer of vaccine antigens or the transfer of immune-boosting molecules, or in some cases, both. The best examples of this group are liposomes and virosomes. AS03 and AS01 can also be mentioned in the category of combined adjuvants [26,27,28]. There are various types of adjuvants with specific characteristics and roles in immunogenicity, some of which are presented below as well as in Table 1.

#### 2.3.1. Mineral Adjuvants

Alum: Aluminum salts were developed in the United States in the 1920s and were the only adjuvant available in the United States for more than seven years until the MF59 adjuvant was approved in the influenza vaccine in the 1990s [29]. Alum is still an important component in most approved human vaccines, such as those for the human papilloma virus (HPV), hepatitis A virus (HAV), diphtheria, hepatitis B virus (HBV), haemophilus, influenza type b (Hib), tetanus, and meningococcal vaccines [30]. Alum has been investigated as an adjuvant in the formulation of the Corona vaccine in several preclinical cases [30]. Aluminum compounds as adjuvants are used for calves and pregnant cattle against bovine herpes virus 1 (IBRV), bovine viral diarrhea virus type 1 (BVDV), parainfluenza virus type 3 (PI3V), and Mannheimia (Pasteurella) haemolytica in veterinary [31].

The results of Liang et al.’s research showed that the formulation of alum with protein S or receptor binding domain (RBD) significantly increased the titer of IgG1 in the serum and the affinity of neutralizing antibodies. Furthermore, B cells induce long-term memory in mice [32].Various parameters can play a role in the selection and formula of adjuvants, such as the type of disease, the way of vaccination, the vaccine platform, the physical and chemical nature of the antigen, the type of immune response required, and the age of the target population. It should be noted that choosing the wrong adjuvant can reduce the effectiveness of the vaccine. Therefore, the selection of vaccine antigens should consider increasing the effectiveness of adjuvants [30].

Calcium phosphate is one of the most important mineral salt adjuvants after aluminum hydroxide, used in diphtheria, tetanus, and pertussis vaccines. This adjuvant increases the IgE antibody response to a lesser extent compared to alum and, in turn, increases IgG1 production [33]. By changing the ratio of IgG1: IgG2α, this adjuvant leads to an immune response via the Th1 pathway [34]. One of the biocompatibility adjuvants is zinc oxide, which can be used and, similar to alum adjuvants, stimulates the Th2 immune response. Cobalt oxide is another adjuvant that has been used. This adjuvant increases the production of IgG2α and reduces the production of IgE, thus reducing the risk of allergic responses to the antigen when used as an adjuvant. In addition, it reduces the induction of lymphatic inflammation [35]. Other metal salts, such as iron oxides and tin, have been less studied as adjuvants. The adjuvant properties of gold and silver NPs have also recently been considered [36].

#### 2.3.2. Emulsion Adjuvants

The history of the use of emulsion adjuvants dates back to 1930 A.D. These emulsions have two-phase systems that require surfactants to stabilize oil in water and fall into several categories, including oil-in-water (O/W), water-in-oil (W/O), or multiple emulsions such as water-in-oil emulsions. In water, they are classified as oils [37]. Water-in-oil emulsions are classified into CFA and IFA adjuvants [38], which are used in some veterinary vaccines, including FMD, equine influenza virus, plague, rabies, influenza, Newcastle disease, and canine hepatitis. Several emulsions of water in oil (W/O) and oil in water (O/W) with mineral oils are used in the vaccination of farm animals against FMD and Newcastle disease [8]. W/O emulsions effectively reduce the cost of vaccine production by reducing the dose or amount of antigen. For example, broilers are vaccinated against Newcastle disease with mineral oil-based adjuvants even if they have only received 1.100 doses of the vaccine; they are protected against the disease [39].

Alternatives with several formulations for CFA and IFA have been identified, the most important of which are Montanide VG 51 ISA, Montanide VG 720 ISA, and Adjuvant 65. Water emulsion adjuvants in 720 ISA Montanide oil and 51 are similar to IFA and have therapeutic applications in cancer, malaria, AIDS, and other autoimmune diseases. Adjuvant 65, another emulsion adjuvant, is the main ingredient in peanut oil and will never be widely used due to peanut allergy concerns. However, it will reduce the dose of antigen required. The MF59 adjuvant, used extensively in veterinary medicine, is an oil-in-water emulsion [8]. The main components of squalene oil are non-ionic surfactants (80 Tween and 85 Span), with identical particles approximately equal to 160. The required gene produces extensive immunogenicity, and its safety has been proven to be used in humans in recent years [40]. It creates an excellent immune environment inside the muscle, which increases the specific immune responses to the antigen. Other studies suggest that it may act, directly or indirectly, as an antigen delivery system, increase phagocytosis and pinocytosis, and increase APC antigen uptake [41]. In addition, compared to MF59 with alum, research has shown that it activates immune cells to a greater extent and increases vaccine antigen use at the injection site. MF59 also increases the number of antigen-presenting APCs in draining lymph nodes (DLN) compared to alum or adjuvant-free vaccines [42].

Other emulsions include AS03, which is based on squalene oil and in which the surfactant 85 Span is absent and contains alpha-tocopherol. Alpha-tocopherol is known as an immune booster because of its high bioavailability. In addition, it strengthens cellular and humoral immunity [38]. Antigen-specific memory cells become DLN. A combined Th2/Th1 response has also been reported with AS03. In AS03, unlike MF59, none of the compounds alone are immunogenic. AS03 contains alpha-tocopherol, which alone strengthens the immune system, and in emulsion additives, boosters of the immune system can enhance the immune response [43,44]. MF59 and AS03 (a similar squalene-based oil-in-water emulsion) have been licensed and used for H1N1 influenza vaccine [45].

#### 2.3.3. Polymeric Adjuvants

Polymeric adjuvants can play an influential role in improving the performance of vaccines by presenting antigens as immune boosters. These compounds are good candidates for preventing pathogenic inflammatory diseases or cancers. Recently, the use of polymer adjuvants in the study of pathogenic mechanisms associated with autoimmune diseases has also been considered [33]. The performance of these adjuvant polymers depends mainly on their solubility, molecular weight, number of branches, and composition of the polymeric scaffold. Polymers can be used in conjunction with live viral and DNA vaccines. Moreover, they are an excellent alternative to conventional adjuvants because of some characteristics, such as biocompatibility and compatibility for formulation with live vaccines expressed by bacterial or viral vectors, high biodegradation, safety, non-toxicity, and ease of production and purification. In the past decades, natural biodegradable polymers of plant and microbial origin and synthetic polymers have been used to improve antigens’ delivery, reducing the reminders needed for vaccines. The natural polymers include fructan, glucan, mannan, chitosan, lipoarabinomanan, muramyl dipeptide, LPS, and Dimycolate Trehalase. Synthetic emulsion polymers include: polyphosphates, polyelectrolytes, polyanhydride, non-ionic block copolymers, polymethacrylates, co polyglycolic lactides, polycaprolactones, and polyvinyl pyrrolidone. Some carbohydrates have adjuvant properties due to their essential signaling roles in the immune system [46].

#### 2.3.4. Saponins

Tripeptide glycosides, or saponins, which are extracted from plants and often from the bark of the Gualaya Saponaria tree, are a group of adjuvants that are both immune and stimulant. Saponins have been widely used as adjuvants for years in several livestock vaccines, including equine influenza virus, canine parvovirus, and febrile leukemia (FeLV). Saponins appear to act more by inducing cytokines. Quil A, isolated by reverse phase chromatography from QS-21 extract, is one of the saponins that stimulates strong cellular responses against antigens and acts as a potent adjuvant to induce Cytotoxic T lymphocytes (CTLs) and produce cytokines type 1 (interleukin and interferon-) and IgG2a isotype antibodies. ISCOMs are particles containing cholesterol, phospholipids, and cell membrane antigens. Quil A lipid, an immune system stimulating compound, is used to reduce the hemolytic activity of saponins. The binding of Quil A to cholesterol prevents its interaction with the cell membrane [1].

ISCOMs are cage-like structures 30–40 nm in diameter composed of glycosides found in cholesterol, Quil A, antigens, and phospholipids [47]. ISCOMs were found to induce a wide range of cellular and humoral immunity [30]. Moreover, hydrophobic or amphipathic antigens can be located within this complex. In addition, they can modulate the immune system and deliver antigens to B cells to increase the rate of antigen uptake by APC. ISCOMs are found in vaccines for bovine viral diarrhea, bovine herpesvirus type 1, bovine plague, FeLV, pseudorabies, and rabies. Sometimes, they have been shown to have toxic effects in mice [48,49].

#### 2.3.5. Derivatives of the Complement System

The components of the mammalian complement system appear to be effective adjuvants to stimulate antibody responses because fragments of this protein bind to foreign antigens and signal them to antibody receptors and immune cells. Regarding component C3d, in one study, the binding of three molecules of C3d to an antigen increased its immunogenicity by up to 1000-fold, and it could be recognized as a useful and special adjuvant [48,49]. Influenza virus hemagglutinin antigens, anti-idiotype antibodies, and capsular polysaccharides have been successfully modified by C3d [1].

#### 2.3.6. Cytokine

Today, cytokine proteins and their genes are used as adjuvants in vaccines. IL-2, IL-1, INF-, granulocyte-macrophage colony-stimulating factors (CSF 3-GM), and IL-12 are examples of cytokines. INF- is a pleiotropic cytokine that can boost cellular immune responses via multiple mechanisms. Granulocyte-macrophage colony-stimulating factor enhances the primary immune response by activating and capturing antigen-supplying cells. In a limited number of experiments, 1-IL and 2-IL-recombinant with other adjuvants were promising for use as adjuvants in cattle and sheep. At present, there is a particular tendency towards IL-12 as the immune response pathway seems to change towards type 1 responses. IL-12 is used in cats as an adjuvant for immunodeficiency virus vaccine subunits [50,51,52]. Cytokines are used in vaccines for various viral diseases of poultry, cattle and fish species. As examples, we can mention IL-18 in the Newcastle disease vaccine and IL-7 in the DNA vaccine for Burs of Fabricius infectious disease. Bovine IL-18 was used as an adjuvant in a DNA vaccine against FMD, which elicited cellular and antibody responses in cattle. Moreover, the use of a DNA vaccine consisting of salmon anemia virus protein and fish type I interferon induced higher levels of antibodies and influx of B cells and CD8 T cells compared to DNA vaccine without cytokines in Atlantic salmon [31].

#### 2.3.7. Adjuvants Taken from Bacteria

The first recombinant vaccine expression systems were developed using E. coli bacteria. The advantage of this system is that it can produce large amounts of defined proteins. However, since prokaryotic cells have different processing mechanisms, expressed proteins are often misfolded. In addition, signal sequences, glycosylation sites, and disulfide bonds, which are present in many candidate vaccine proteins, can result in toxicity, insolubility, or rapid degradation in bacteria [53]. Their activity is affected by the activation of TLR-like receptors through hazard signals of the host defense system. As a result, heat-killed and lyophilized products containing Propionibacterium avidum KP-40 can increase the antibody response to the antigen that is distributed with them. In the immunomodulatory mycobacterium peptidoglycans, muramyl dipeptide (MDP) is an active compound. It has important side effects, including fever, arthritis, and iris inflammation. Nevertheless, it produces fewer toxic derivatives. Hydrophilic compounds stimulate most type 2 immune responses, including the use of threonyl-MDP in FeLV vaccines, whereas hydrophobic derivatives are often incorporated into compounds, such as liposomes or ethanolamines in water or oil emulsions, which are used to stimulate type 1 reactions and cellular immunity [1,50].

**Table 1 vaccines-11-00453-t001:** Types of adjuvants and their roles.

Adjuvants	Role	References
Mineral adjuvants	increase IgE and IgG1 changing the ratio of IgG1: IgG2α, lead to an immune response to the Th1 pathway	[33,34]
2. Emulsion adjuvants	increase phagocytosis and pinocytosis increase APC antigen uptake. Consequently, increases the number of APCs in DLN	[41,42]
3. Polymeric adjuvants	improving the performance of vaccines by presenting antigens as immune boosters. improve antigens’ delivery, reducing the dose of reminders needed for vaccines.	[33,46]
4. Saponins	stimulating cellular responses, induce CTL produce cytokines type 1 (interleukin and interferon-γ) and IgG2a isotype antibodies	[1]
5. Derivatives of the complement system	stimulate antibody responses	[48]
6. Cytokine	enhance cellular immune responses and seems to change towards type 1 responses	[50,51,52]
7. Adjuvants taken from bacteria	Their activity is affected by the activation of TLR-like receptors through hazard signals of the host defense system	[1,50]

### 2.4. Investigatation the Side Effects of Adjuvant Use

The potential and toxicity of adjuvants must be balanced to provide the safest stimulation with the fewest side effects. To use adjuvants, pay attention to how they are used. This is a fundamental issue. For example, there are unacceptable adverse effects for human use in the preparation of veterinary vaccines. In livestock, several side effects that affect the animal’s growth, reproduction rate, or well-being have an effect or cause. Vaccines used for prophylaxis in healthy people are needed to prevent side effects, although they are less critical in therapeutic vaccines to treat diseases, such as cancer and AIDS. Another critical factor is the age of the individual. The use of vaccines for diseases transmitted from the mother has recently been studied in pregnant women. Scientists have turned their attention to the teratogenic effects of new additives [48,50]. The inoculation method may also be effective against the side effects of adjuvants. Regional burning, itching, erythema, and pain may be observed with the subcutaneous method. In addition, transient swelling may occur due to inflammation as the vaccine enters the sensory nerve cell compartment. Minor swelling is created in the intramuscular method because the area is more profound and does not have sensory neurons [50].

Intraperitoneal injection of adjuvants in animals can lead to chemical peritoneal inflammation, ascites, and fibrosis formation between various organs in the body. The reactions that usually occur are regional tenderness and swelling. Furthermore, more severe reactions, such as the formation of painful puddles and nodules, are observed in animal models, though weight loss and piloerection are also observed. As a result of using live vaccines with adjuvants, we have an increase in body temperature, which reduces or interrupts feeding and reduces milk production (in the case of breastfeeding). Other known side effects in animals include transient edema and fever, increased saliva, vomiting, diarrhea, urticaria, arthritis, uveitis, discoloration of the skin at the injection site, anorexia nervosa, nausea, injury, fever, and abortion, which has also been reported. Side effects are rare, depending on the dose of vaccine used annually [33].

Killed vaccines are as safe as live attenuated vaccines, with mainly regional side effects. The reaction at the injection site is a significant concern in breeding animals. Oil-based adjuvants are used primarily in veterinary vaccines; they may induce local reactions such as granulomas and fever. Local wounds and tissue necrosis are possible by inoculation with mineral oil emulsions due to the short-chain hydrocarbons in their combination. High-purity non-mineral oils are much more tolerable. They are rapidly metabolized and eliminated from the injection site, and only transient, weak local inflammation occurs. In contrast, mineral oils remain at the injection site and are either removed by cells such as macrophages or decomposed to a lesser extent into fatty acids, triglycerides, phospholipids, or sterols [54,55,56].

Systemic side effects that may occur due to overstimulation of the immune system due to immunosuppressive drugs and vaccines include acute-phase responses, allergies, autoimmune disease, and autoimmune disease [57]. Although alum adjuvant has long been used in licensed vaccines, investigating its adverse effects is of great importance. Aluminum is classified as a neurotoxin and may cause some neurological disorders, including ADS, autism spectrum disorder, Alzheimer’s, and amyotrophic lateral sclerosis (ALS) [58].

Compared to the health benefits of using alum as an adjuvant, its side effects are much less. Be aware, however, that alum is the only adjuvant that has been licensed for use in vaccines for nearly a century, and its removal from the world will have far-reaching adverse effects [29]. Concerns have also been raised about the safety and tolerability of AS03 vaccines which have this adjuvant. In some studies, the development of drowsiness is indicated in children [59].

The number of currently approved adjuvants is minimal, and in the United States, only five adjuvants have been approved: alum, AS04, MF59, AS01, and CpG 1018. Meanwhile, alum has been approved as an adjuvant for human use [36,60].

## 3. Role of Adjuvants in Immunogenicity

### 3.1. Adjuvant Mechanism in Immunogenicity

According to the studies, the mechanism of action of an adjuvant has been identified only to the extent that it generally stabilizes and increases the supply of antigens or acts as a modulator of the immune system. An adjuvant may have more than one mechanism of action. For example, adjuvants that help maintain the structure of the antigen at the same time can also increase the vaccine’s quality and shelf life. Antigens that affect antigen delivery can affect countless points in this complex process. Antigens are often carried to lymph nodes during an immune response by dendritic cells. The antigen-APCs include dendritic cells, macrophages, and B lymphocytes, which process antigens and deliver epitopes to T cells in the form of MHCs. APCs provide signals to initiate an immune response, such as auxiliary stimulation by B7 family molecules. Any adjuvant that increases antigen uptake by these cells increases the expression of Major histocompatibility complex (MHC) molecules and improves the immune response by increasing cell migration to the lymph nodes. Some adjuvants reduce the liver’s effect on the gene by trapping and providing a continuous supply of antigens to antigen-supplying cells locally at the injection site. New microparticle adjuvants can create long-term (1–6 month) depots and gradually release a certain number of other adjuvants that may be effective by saturating Kupffer cells in the liver and reducing hepatic antigen uptake. The reactions are activated by the interaction of helper T cells and CTL with antigens present in MHCI or MHCII molecules. Although antigens are presented differently, antigens on MHCII molecules are generally absorbed outside the APC cells by phagocytosis. However, antigens in MHCI molecules are from the cytoplasm of all cells supplying this molecule. Most adjuvants can effectively stimulate helper T cells and humoral immunity [50].

The results of one study showed that APC activation occurs when pattern recognition receptors on the surface of antigen-supplying cells attach to protected areas in bacterial lipopolysaccharides, carbohydrates, or other parts. If this hypothesis is correct, adjuvants can mimic the original signals. In fact, many adjuvants are bacterial derivatives or similar compounds of proteins, carbohydrates, or bacterial DNA. However, this model is not accepted for oil emulsion additives, saponins, or alum. According to the second model, APCs act by detecting internal signals released by damaged cells, stressed cells, and dying cells. According to research, necrotic fibroblasts or blood vessels can act as highly effective adjuvants and affect the function of neutrophils after vaccination with some adjuvants. Adjuvants, such as liposomes, deliver antigens to pathways that lead to their expression in MHCI molecules and produce the CTL response. Cross-presentation of antigens from two different pathways may sometimes be important in the production of CTLs. Another mechanism of activity of adjuvants is the effect on the immune system, which is the mechanism of modulating the immune system and stimulating the cytokine network of the immune system. Increasing and decreasing the concentration of cytokines affect the type of immune response. Cytokines, such as interferon-gamma (IFN-γ), IL-2, and IL-12, are associated with type I helper T cell (THI) responses and cellular immunity, and cytokines IL-4, IL-5, IL-6, IL13, and IL-10 may be associated with TH2 responses and humoral immunity [50].

There is a hypothesis related to the activity of the NLRP3 (NALP3) inflammatory receptor. An intracellular pattern recognition (PRR) receptor of the NLR family is activated during the innate immune response to recognize bacterial (such as lipopolysaccharide) compounds, zymogenesis, and toxins, to announce stress and cell damage to other members of the immune system. After oligomerization of NLRP3 protein units, the activity of the inflammatory complex begins, and then the enzyme caspase1 is activated to stimulate the proteolytic breakdown of precursor molecules. This complex eventually triggers the production of immune responses by inflammatory cytokines of the IL-1 family, including IL-1β and IL-18 [33,61].

### 3.2. Immunogenicity Mechanism of Multiple Adjuvants

Aluminum adjuvants elicit strong innate immune responses that include the invasion of macrophages, neutrophils, mast cells, natural killer cells, CD11b+ monocytes, and dendritic (DC) cells [42,62]. Upon entry of CD11b+ monocytes, they are differentiated into inflammatory dendritic cells at the injection site [63]. These cells are essential for the activity of alum as an adjuvant. In addition to their role in creating an innate immune response, granulocytes trigger an acquired Th2 response and the production of IgM antibodies. After stimulation with alum adjuvant, primary macrophages produce E2 (PGF2α) and IL-1β (IL-6). PGF2α can stimulate the Th2 response, and thus the T cell response is effective in injecting induced alum. On the other hand, PGF2α has a direct effect on B cells and increases IgE production [64,65,66].

These particulate adjuvants increase mass formation to facilitate phagocytosis. Carbohydrate polymers such as mannan can bind antigens to antigen-supplying cells by binding to carbohydrate receptors. Carrier proteins such as bovine serum albumin (BSA) (KLH) and diphtheria or tetanus toxoid provide haptens or carbohydrate antigens through T cell uptake. Some adjuvants also appear to stimulate the cytotoxic T lymphocyte (CTL) response by placing antigens in specific parts of antigen-supplying cells [50].

MF59 water-based oil adjuvant activates more immune cells than alum, resulting in a higher injection of vaccine antigen at the injection site. Moreover, MF59 increases the number of APCs in DLN compared to alum or adjuvant-free vaccines [42].

Some studies link the apoptosis-associated Speck-like effects of MF59-dependent proteins to ASC (CARD), common moderators of inflammasome complexes, while others mention an MF59 adjuvant effect independent of NLRP3 and Caspase1 inflammasomes [67]. Therefore, it is possible that ASC has a function independent of the inflammasome or has a role in the inflammasome other than NLRP3 [68].

Saponins may stimulate cellular immunity to antigens by altering the levels of these two classes of cytokines, although they naturally only stimulate the production of antibodies. Several immune-stimulating adjuvants increase the expression of MHC molecules on APC cells, either directly or by producing cytokines. According to some hypotheses, the APC must first activate and initiate an immune response [50].

## 4. Nano-Adjuvants

### 4.1. Properties of Nano-Adjuvants

Nanoparticles’ sizes range from 1 to 1000 nanometers (1 micrometer) [69]. MPs are defined as particles with sizes between 1 and 1000 μm [70]. MPs stimulate humoral immune responses, while NPs tend to induce cellular immune responses [71].

In order to overcome the limitations of adjuvants and the effective and controlled delivery of antigens, attention has been drawn to nano-carriers that can be a promising platform for presenting and stimulating a better immune system. The introduction of TLR-specific agonists into nano-polymers resulted in better absorption of APCs and a longer T cell response. Uploading antigens to CPG-binding nano-polymers activates DCs, and NPs can be used to deliver antigens to lymph nodes [72,73].

In next-generation vaccines, nano-adjuvants with a wide range of compounds, such as minerals and organic matter, microorganism-derived components, emulsions, microorganism-derived components, and various molecules in amplification and shaping safety response are involved [74]. Some of the polymers used in the preparation of NPs are: lactic polyacids, ortho polyesters, PLGA, and natural polymers, such as albumin, collagen, gelatin, chitosan, and alginates, which are approved for clinical use in humans because they are biodegradable and non-toxic [75,76]. (PLGA) has been used for decades as a constituent of NPs due to its excellent compatibility, biodegradability, and safety profile. The Food and Drug Administration has also approved its use in humans [5,77,78]. PLGA is hydrolyzed by the body to produce lactic acid and glycolic acid. Moreover, they are effectively metabolized through the Krebs cycle, so they have no toxic effect. They are also used to produce tellurogenic vaccines for autoimmune diseases, including experimental autoimmune encephalomyelitis (EAE) and an animal model of multiple sclerosis. PLGA has been shown to be an excellent biocompatible polymer for NPs because it has an excellent ability to combine with a wide range of molecules and improve their transport to target tissues. PLGA is used in protein-based viral vaccines, DNA vaccines, mRNA vaccines, and inverse vaccines. The reported results showed that PLGA NPs could enhance the immune response even without loading [77].

There are several methods for producing Nano-adjuvants, which, regardless of the particle-making method, provide important factors such as size, surface charge, and formulation because their intracellular function is complex and affects their performance [11,79]. There are different data points on the relationship between particle size and the quality of the immune response produced. Some studies show NPs have a more remarkable ability to act as adjuvants than microparticles [11,80]. Small particles are more efficient at penetrating biological barriers and are better distributed in bloodstream than larger ones. NPs with a diameter of 100 nm or less are preferred over larger particles for drug delivery purposes. Furthermore, small NPs in the size range of 20–200 nm can rapidly drain into the lymph nodes (LN) and be taken up by resident DCs. Large NPs from 500 to 1000 nm migrate from the injection site (skin) to the LN in vivo by cellular transport by DCs. These data suggest that larger NPs preferentially interact with tissue-resident APCs, while smaller NPs (<200 nm) provide better antigen presentation because they can circulate through a vein and lymphatic drainage [81].

### 4.2. Most Common Types of Nano-Adjuvants

Nano-adjuvants can also be categorized into different types depending on the type of NP. In addition, they show different roles in immunogenicity. Table 2 lists the various types of nanoadjuvants, as well as their specific roles and applications in vaccines.

#### 4.2.1. Gold NPs

Gold (Au) has been widely used in nanomedicine due to its therapeutic effects on several diseases in the form of NPs. Gold can also play an important role in the development of vaccines as an adjuvant, increasing the immunogenicity of antigens, and reducing toxicity and stability (Table 2) [82]. Gold NPs with chitosan (CsAuNPs) were used as carriers for tetanus toxoid (TT) as a model antigen, along with the immune stimulant Quillaja Saponaria (QS) as an adjuvant. Oral immunization of mice with CsAuNPs-TT-QS induced immune responses up to 28 times. Therefore, combining adjuvants with NPs can play an important role in the effectiveness and stability of vaccines, especially mucosal vaccines [30]. Metal NPs have rigid structures, and at the same time, their synthesis is simple and relatively non-biodegradable. At present, for the synthesis of metal NPs, biological materials, such as bacteria, plants, and algae, are usually used [83].

#### 4.2.2. Silver NPs

Ag NP is one of the most vital and attractive nanomaterials among several metal NPs, which have been exponentially used as antimicrobial and larvicidal agents due to many advantages, such as low production cost and simplicity of synthesis. Recently, several studies have been conducted to use synthetic NPs as adjuvants to enhance the immunogenicity of antigens [83]. The silver NPs increased the immune response against the inactivated rabies virus in a mouse model, and increased the potency of veterinary rabies vaccine without in vivo toxicity compared to commercial alum adjuvant [84].

#### 4.2.3. Polylactic-Co-Glycolic Acid

The most common synthetic biodegradable polymers are NPs as PLGA adjuvants. PLGA is a highly compatible copolymer composed of polylactic acid (PLA) and polyglycolic acid [30]. PLGANPs loaded with a wide variety of molecules are approved by the US Food and Drug Administration (FDA) as carriers for human and veterinary drug use [85]. The slow release mechanism of antigens and adjuvants in PLGA vaccine/adjuvant capsulation can stimulate innate immune responses, potentially triggering mucosal and systemic immune responses and improving humoral immune memory in CD8+ T cells [30]. In addition to capsulation, antigens can be adsorbed on the PLGA surface by electrostatic or hydrophobic interactions [86]. The newly developed PLGA-ICMV platform can be considered a promising candidate for carrying therapeutic vaccines because it has excellent potential for encapsulating water-soluble biological agents, such as proteins and DNA plasmids [87].

#### 4.2.4. Chitosan

Chitosan showed promising mucoabsorption effects due to high mucoadhesion and the transient opening of mucosal cell membrane tight junctions. The interaction between the positive charge of chitosan and the negative charge of mucin can increase the contact time between the drug and the absorption surface and be used in mucosal vaccines [88]. For example, co-application of chitosan to mucus and live attenuated Newcastle disease virus enhanced Th1 cell-mediated immune responses in chickens. Moreover, intranasal immunization of cows with chitosan-adjuvanted FMD vaccine protected against direct exposure to FMDV [31].

Recently, Renu et al. (2020) developed a subunit chitosan NP-based vaccine using immunogenic outer membrane proteins (OMPs) and flagellin (F) protein (OMPs-F-CS NPs) of Salmonella. It significantly increased lymphocyte proliferation and reduced salmonellosis in poultry [89].

#### 4.2.5. Liposomes

Liposomes are the most commonly investigated nanostructure systems used in advanced drug delivery, which were first discovered by Bangham in 1963 and Horne introduced liposome technology as a system for ion diffusion across biological membranes, and later in the 1970s, interest in therapeutic applications developed [90,91,92]. They can take over the natural structure of the cell membrane and have long been investigated as drug carriers due to their excellent absorption capacity, safety, and biocompatibility [93]. Liposomes are spherical vesicles characterized by a bilayer of phospholipids with an internal aqueous cavity. Injectable forms of liposomes as carriers have been licensed for clinical use [94]. As a biodegradable biopolymer, chitosan is used to coat liposomes to improve their effectiveness. Liposome-based vaccines are used in veterinary medicine as oral carriers to target a wide range of viral and bacterial diseases, such as Salmonella enteritidis. In addition to traditional liposomes, bilosomes are proposed, which are lipid-based non-ionic vesicles (niosomes) containing biodegradable and biocompatible bile salts (sodium deoxycholate), which due to their adjuvant properties and rapid uptake by M cells have been widely used for oral vaccine [30].

**Table 2 vaccines-11-00453-t002:** Common types of nano-adjuvants and their role in immunogenicity.

Nano-Adjuvants	Role in Immunogenicity	Example of Application in Vaccines	References
PLGA	can stimulate innate immune responses by a slow-release mechanism of antigens and adjuvants improvement of humoral immune memory CD8+ T cells	Influenza A vaccine hepatitis B virus (HBV) vaccine	[5,30]
2. Chitosan	charge of chitosan and the negative charge of mucin can increase the contact time between the drug and the absorption surface stimulating IgG IgA and μ-IFN responses	(OMPs-F-CS NPs) of Salmonella vaccine	[87,88,89,95,96]
3. AuNPs	increasing the immunogenicity of antigens, and reducing toxicity and stability induced immune responses up to 28 times	(CsAuNPs) were used as a carrier for tetanus toxoid (TT) vaccine	[30,82]
4. Liposome	Rapid uptake by M cells	oral carriers to viral and bacterial diseases, such as Salmonella enteritidis	[30]

### 4.3. The Impact of Nano-Adjuvants on the Immune System Response

To induce an immune response, the path of the antigen reaching the lymphatic tissues is so important that different vaccination methods can affect the induction of the immune response. When NPs are actively targeted, such as through ligand binding, their uptake and distribution may change significantly. This particular interaction between the target ligand and some tissues enhances the accumulation of NPs at a specific location [97,98], which improves tissue binding and absorption, which is very effective in mucosal immunization, especially in human inhalation vaccines. According to the results, NPs easily cross the existing extracellular barriers, while large particles are rarely transported to places other than the lungs and cleared by mucosal movements or phagocytosis [99].

In nano-adjuvants, larger particles adhere better to APC surfaces and act as a repository for the continuous release of antigens. On the other hand, NPs perform better in targeted antigen delivery because they cross biological barriers more easily. However, particles will not interact when they are much larger than APCs [100].

According to some research, particle size can affect immune responses. Microparticles induce a more humoral immune response, and NPs strengthen the cellular immune response. DCs absorb particles with a diameter of 500 nm or more through phagocytosis and pinocytosis, which leads to a humoral immune response. Particles of 20–200 nm mainly produce immune responses through TH1L by endocytosis and subsequent stimulation of CD4+ and CD8+. Thus, it is argued that NPs better absorb and stimulate cellular immune responses by APCs than microparticles, as evidenced by test results on AIDS and hepatitis vaccines. In contrast, some results from a study on influenza virus antigen and bovine serum albumin loaded on PLGA have made it difficult to classify and accurately predict the particle size range affecting TH1/TH2 responses. Therefore, better antigen delivery by nano-adjuvants depends on other factors such as the degree of hydrophobicity of the NPs’ surface, the charge, and the type of peptide/ligand [68,72].

### 4.4. Application of Nano-Adjuvants

Body mucosa is one of the major entry points for pathogens, so mucosal immunization can overcome the limitations of injectable immunization. However, unfortunately, the number of licensed mucosal vaccines is low due to the inefficiency of antigen delivery systems. Today, nano-adjuvants are designed to improve the delivery system of antigens, which can simultaneously deliver several stimulatory components of the immune system to achieve a synergistic effect [95,101,102].

Nano-adjuvants that inactively target APCs are better absorbed than free antigens. On the other hand, coating them with FDA-approved biodegradable PLGA can be very effective preventing proteolytic degradation of NP. Aside from the targeted delivery of antigen to APCs, another advantage of nano-adjuvants is the ease of chemical changes to their surface, such as changes in surface charge or hydrophobicity, so that a balance of adhesion and strength can support specific immune responses against the antigen [96,101].

Cellulose derivatives, chitosan, alginate, and polyvinyl alcohol cause better distribution of antigens throughout the mucosal tissues due to their hydrophobic properties by preventing the accumulation of particles in the mucosa. Chitosan has been studied in mucosal vaccination. Chitosan is a cationic molecule with high interaction power with negative mucosa to enhance adhesion. Opening epithelial interfaces increase antigen penetration, has adjuvant activity and causes stimulating IgG IgA and μ-IFN responses. Chitosan particles with smaller dimensions are more active and have no destructive or toxic effects. In mucosal vaccines based on chitosan adjuvants, TLR4, CD14, and mannose receptors detect this polysaccharide for innate immunity. Chitosan induces the TH1 response via the DNA cGAS-STING pathway [80,95,96].

The major APCs in the primary immune response are DCs, and activating PRPs by their ligands causes DCs to mature. They also produce pro-inflammatory cytokines that regulate the immune response. Therefore, ideal nano-adjuvants should, in addition to actively targeting APCs, protect against TLR antigens and ligands to induce efficient maturation in DCs. Subsequently, mature DCs promote B-cell receptors by delivering antigens to the cell surface and sequentially interacting with THs to secrete antibodies. As a result, the ligand binds to DC membrane receptors [103].

NPs provide profiles for the secretion of chemokines, cytokines, and the stable secretion of antigens prior to uptake by APCs. After APC activation due to exposure to nano-adjuvants, these molecules activate MHC and excitatory molecules. Cells that absorb antigens by phagocytosis and endocytosis convert these antigens into small peptides. Moreover, they exhibit this through MHCs at their surface, and by creating stimulus molecules, they have a synergistic effect on activating antigen-specific T cells. After these steps, adult APCs migrate to the lymph nodes to bind to specific antigens [104].

TLRs are internal messages that specifically affect DCs and subsequently increase antigen delivery to APCs, directing them to the MHCI and MHCII molecules, where more interferon and cytokines are produced. Then, by stimulating TLRs on DC and B cells, synergistic antibody responses were observed [105,106]. Due to intracellular localization, processed APCs deliver antigens in the cytosol through MHCI and MHCII molecules. In addition, PRRs in different cell parts are involved in identifying pathogens and acting as stimuli for the T cell response [107]. mRNA patterns for the expression of TLRs are different in tissues and immune cells. For example, in most tissues, TLR1/6/10, TLR6, TLR3, and TLR5 are expressed, but immune cells, such as DCs, macrophages, neutrophils, and cell subsets B and T, only express the TLR 2 and TLR7 genes [105].

Therefore, to modulate specific responses against an antigen, the intracellular location of nano-adjuvants must be regulated by directing PAMPs that affect the activation of the PRR message pathway [108]. Despite the brilliant performance of nanotechnology and its applications and the formation of some promising vaccine strategies, there are still many concerns and issues to consider and in order to overcome the fundamental challenges of optimizing biological behaviors and reducing the potential risks of nanomaterials. Given the ability to optimally transfer NPs from biological barriers, such as BBB transfer and embryo transfer, we must evaluate it fully before widespread clinical use [11].

### 4.5. Factors Influencing the Adjuvant Efficacy of NPs

#### 4.5.1. Hydrophilic and Hydrophobic Features of NPs Surfaces

Concerns about the health of NPs focus on the biological fate of NPs and the accumulation of some non-degradable substances in the body. Under these conditions, NPs can persist for several months in macrophages and stromal cells, leading to toxicity. Hydrophobic NPs are generally unstable and disperse in biological fluids and culture media. Particle–particle hydrophobic interactions in the biological environment increase the accumulation of hydrophilic particles [76,79].

#### 4.5.2. Particle Charge and Coating Composition

Adjuvant NPs are typically used for therapeutic or diagnostic purposes. In this case, a series of measures must be taken, which in most cases include the loading of NPs with specific biological molecules such as peptides, ligands or chemical groups. The aim is to increase the affinity between the nano-adjuvants’ surface molecules and the membrane receptor [101].

Antigen uptake by nano-adjuvants is based on the charge or hydrophobic interactions between the candidate molecule and the NPs. At this stage, the bonds formed are mostly non-covalent. The separation of the antigen from the nano-adjuvant depends on environmental conditions, such as pH, temperature, ionic strength, and antigen hydrophobicity. For targeted antigen delivery, most cationic groups perform better. Due to electrostatic interactions between cationic NPs and anion cell membranes, they adhere more easily to the cell surface. According to the results, the amount and frequency of surface cell uptake increase with an increased positive charge. In contrast, larger particles with a high positive charge may cause toxic effects by creating pores in the membrane [109]. In addition to the charge and size of NPs, the shape of these nanoagents is also a determining factor in antigen release, cellular interactions, and intracellular traffic in the host. Spherical nano-adjuvants, for example, complete the immune response faster than cubic and rod nano-adjuvants [79].

## 5. Conclusions

Based on studies, vaccination is the best way to prevent and reduce the damage caused by infectious diseases in animals and humans. Therefore, they are considered for the treatment and prevention of diseases, considering the basic need for adjuvants in veterinary vaccines. Recently, efforts have been made to reduce the complexity of antigens, such as pure antigens, recombinants, synthetic proteins, or even synthetic peptides, as adjuvants. In recent years, NPs have received a lot of attention and been used as adjuvants. They have significant advantages, such as the targeted delivery of antigens to APCs, ease of chemical changes on their surface, such as changes in surface charge or hydrophobicity, good adhesion and strength, and they create specific immune responses against antigens. They also simultaneously provide multiple immune stimulatory components to achieve a synergistic effect and provide profiles for the secretion of chemokines, cytokines, and the sustained release of antigens prior to uptake by APCs. After APC activation due to exposure to nano-adjuvants, these molecules activate MHC and stimulatory molecules.

Despite the brilliant performance of nanotechnology and its applications and the formation of some promising vaccine strategies, there are still many issues to overcome in optimizing biological behaviors and reducing the potential risks of adjuvants. In addition to the negative effects they can have, such as inflammation, edema, and neurological disorders, there are a number of other concerns that, due to the ability of nanoparticles to be optimally transferred from biological barriers, such as BBB transfer and fetal transfer, should be evaluated before widespread clinical use of adjuvants. On the other hand, the level of immune responses caused by nanomaterials should also be carefully examined to avoid toxic reactions. So far, few studies have been conducted to predict the effects of some properties of nanoparticles. Therefore, more empirical data are needed to establish a standard framework. The need for immediate progress in this field is felt due to the development of nanotechnology for vaccines and their therapeutic applications.

## Figures and Tables

**Figure 1 vaccines-11-00453-f001:**
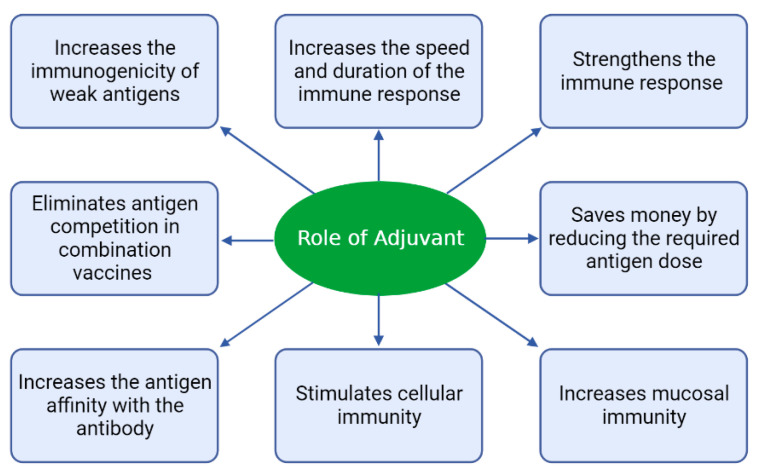
Different ways in which adjuvants play a role in improving immune responses in the body [1].

## Data Availability

No new data were created or analyzed in this study. Data sharing is not applicable to this article.

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
