# Peer review of "Immunogenicity of Different Types of Adjuvants and Nano-Adjuvants in Veterinary Vaccines: A Comprehensive Review"

_vaccines, 2023, doi:10.3390/vaccines11020453_

Round 1

Reviewer 1 Report

The manuscript is a review of adjuvants and nano-adjuvants most commonly used in veterinary vaccines. It is well written, and it is of interest for readers, particularly for those working in vaccine development for the control of different types of pathogens. Some aspects that should be improved are:

1.- The title should be changed since the manuscript is a review, not an “evaluation” of  nano-adjuvant. In addition, more than half of the manuscript is a review of adjuvants not only nano-adjuvant as is stated in the title. This is why a title like “Review of adjuvant and nano-adjuvant immunogenicity in veterinary vaccines” could be more appropriate.

2.- The manuscript is about veterinary vaccines, however, in the review of many adjuvants the authors only stated human vaccines, for example in alum several human vaccines are cited but no veterinary vaccine is cited. This would be important for readers, even vaccines under development showing promising results should be cited in the appropriate adjuvant and nano-adjuvant. The same can be applied to the MF59, AS03, polymeric adjuvants, complement, cytokines….

3.- Tables 1 and 2 should be cited in the text.

4.- References should be revised for appropriate format, some of them are abbreviated and other do not.

Author Response

Dear Reviewer,

Thank you for your precise and useful comments. Our responses to each comment are listed as follows:

  1. You are completely right about the title and thank you for your suggestion. So, it has been changed into the: Immunogenicity of Different Types of Adjuvants and Nano-adjuvants in Veterinary Vaccines: A Comprehensive Review
  2. It been tried to add some examples of mentioned adjuvants in veterinary vaccines in the text.
  3. Table 1 and 2 has been cited in the text.
  4. References have been revised and corrected; now all of them are complete and according to Vancouver

In addition, some grammatical errors have been fixed, and the text has been checked for its English writing.

Thank you again for your time and consideration.

Reviewer 2 Report

S. Nooraei et al. submitted a review upon nano-adjuvants in veterinary vaccines. This manuscript fell within the scope of Vaccines, but some flaws existed. A substantial improvement should be made. It might be reconsidered after a Major Revision. The authors must address the following issues point-to-point.

Q1: The full names of abbreviations must be provided at the first appearance. For instance, please offer the full name of ‘APCs’ in Abstract, ‘NK’ in Introduction, etc.

Q2: The keywords were not informative. ‘Nano-adjuvant’ was a derivate from ‘adjuvant’, and thus only two valid keywords were shown. More keywords must be provided.

Q3: Were the authors sure of that most cited and reviewed papers in the main text were about veterinary vaccines? The reviewer supposed that some of them might be about human vaccines. Of note, if a certain paper was aimed to discuss human vaccines, it might be improper to be used as a material to discuss veterinary vaccines.

Q4: The Introduction Section emphasized the background of vaccines, but exhibited much fewer information about veterinary vaccines. The Title of current work revealed that the focus was veterinary vaccines. Please modify the structure of Introduction, so as to pay more attention to the background of veterinary vaccines.

Q5: In Section 2~4, although some Tables were shown, the reviewer suggested to add some schematic summaries upon some important topics, e.g., the side effects and mechanisms of adjuvants. Vaccines is a flagship journal that published high-quality reviews.

Q6: The definition and types of adjuvants were introduced in Section 2. However, some types of adjuvants herein might be regarded as nano-adjuvants, like liposomes. Actually, there was another section of nano-adjuvant in Section 4. The authors must keep consistent in their review aims.

Q7: Only one subtitle was defined in Section 3. Please consider to define more subtitles to optimize the structure.

Q8: In Section 4.2, metal-based NPs were juxtaposed with PLGA and chitosan NPs. This was inappropriate. Because PLGA and chitosan were concrete materials with defined chemical structure, while metal was a category of several different materials. Similar issues existed in Table 2, where only one kind of NP (CsAuNPs) was taken as an example for metal-based NPs.

Q9: What were the significance of Section 4.4 and 4.5? Did the authors intend to list hydrophobicity, charge and coating as the influencing factors for vaccination efficacy? Then please try to transform them into a section like ‘Influencing factors for vaccination efficacy’.

Q10: To end Section 4 with ‘Advantages of nano-adjuvants’ was to some extent weird. Maybe ‘Application of nano-adjuvants’ would be better, along with the proper modification of the content.

Q11: It was less frequent to cite a number of papers in Conclusion Section. Conclusion allowed the authors to put forward their innovative thoughts.

Q12: The Author Contributions was missing.

Author Response

Dear Reviewer,

Thank you for your precise and useful comments. Our responses to each comment are listed as follows:

  1. As you mentioned all abbreviations are now provided in full name at first appearance in the text.
  2. Two other related keywords have been added.
  3. You are right. Some examples of the use of adjuvants in human vaccines were mentioned in the text. However, it was also tried to add veterinary examples.
  4. The structure of the introduction has been modified for a better reflection of the background of veterinary vaccines.
  5. A figure was added which summarize and mention the different role of adjuvants.
  6. The liposome has been transferred into the types of nano-adjuvants section.
  7. Two subtitles were added in section 3.
  8. That is right. So, it has been corrected.
  9. The title has been changed to Factors influencing the adjuvant efficacy of NPs.
  10. The title of this section has been changed according to your suggestion into the Application of nano-adjuvants.
  11. Conclusion has also revised and modified.
  12. The author contributions have been added.

In addition, some grammatical errors have been fixed, and the text has been checked and improved for its English writing.

Thank you again for your time and consideration.

Round 2

Reviewer 2 Report

The authors had improved their manuscript.